# Inhibiting the Interaction Between Phospholipase A2 and Phospholipid Serine as a Potential Therapeutic Method for Pneumonia

**DOI:** 10.3390/cimb47070516

**Published:** 2025-07-04

**Authors:** Jianyu Wang, Huanchun Xing, Lin Wang, Zhongxing Xu, Xin Sui, Yuan Luo, Jun Yang, Yongan Wang

**Affiliations:** State Key Laboratory of Toxicology and Medical Countermeasures, Beijing Institutes of Pharmacology and Toxicology, Beijing 100850, China; wjy1230666@163.com (J.W.); xinghuanchun0205@163.com (H.X.); 19910725903@163.com (L.W.); 18503826187@163.com (Z.X.); sx_226@163.com (X.S.); luoyuan2006@163.com (Y.L.)

**Keywords:** pneumonia, alveolar surfactant, lipopolysaccharides, phospholipase A2 inhibitor, alveolar macrophages, antibiotic resistance

## Abstract

Pneumonia is a severe lower respiratory tract infection. This study demonstrates that phospholipase A2 (PLA2), a potential biomarker for pneumonia, contributes to alveoli damage by hydrolyzing pulmonary surfactant phospholipids. This process impairs gas exchange and generates hemolytic phospholipids that disrupt cellular membranes, exacerbating pulmonary injury. Experimental evidence demonstrates that PLA2 inhibitors significantly alleviate cellular damage in lipopolysaccharide (LPS)-induced pulmonary inflammation. These findings reveal a key mechanistic role of PLA2 in pneumonia pathogenesis and suggest novel therapeutic strategies. The results may provide more effective clinical interventions and guide further research in related fields.

## 1. Introduction

Pneumonia, a severe lower respiratory tract infection [1], ranks as the fourth leading cause of global mortality [2], accounting for 2.6 million deaths worldwide in 2019 [3]. While predominantly caused by specific bacterial pathogens, the mechanistic details of pulmonary infection remain partially elucidated. Gram-negative bacterial lipopolysaccharides (LPSs) stimulate macrophages activation, triggering an inflammatory response. Notably, this response includes upregulated phospholipase A2 (PLA2) [4], representing a conserved immune defense mechanism against bacterial invasion [5]. However, the precise role of PLA2 requires further clarification. 

PLA2 catalyzes the hydrolysis of ester bond at the sn-2 position of glycerophospholipids, generating free fatty acids and lysophospholipids (ly-PLs). As a key enzymatic mediator, PLA2 plays a vital role in various biological processes, particularly in cell membranes’ phospholipid metabolism, where it regulates critical physiological processes including apoptosis and inflammatory responses. Previous studies have demonstrated that resveratrol suppresses LPS-induced PLA2 expression in rat lung tissue, indicating its potential therapeutic value for pneumonia treatment [6]. However, the precise protective mechanisms and pneumonia-specific functions of PLA2 remain incompletely characterized, warranting further investigation. 

Most reports investigating the PLA2-mediated acceleration of inflammatory processes have primarily focused on the phospholipid (PL) breakdown products, particularly fatty acids, such as arachidonic acid and their activation of cyclooxygenase metabolic pathway [7]. In contrast, the biological role of lys-PLs, another major class of PL metabolites, remains relatively understudied. A pulmonary surfactant, which is rich in PLs and essential for alveolar stability and gas exchange [8], allows a possible distinct interaction with PLA2 through phospholipid hydrolysis [9]. The PLA2-catalyzed generation of ly-PLs from surfactant PLs could significantly compromise pulmonary surfactant function, potentially leading to alveolar instability and impaired lung function. In this study, we aimed to investigate the mechanisms by which PLA2 contributes to inflammatory development in pneumonia and to evaluate the therapeutic potential of PLA2 inhibitors. 

## 2. Materials and Methods

### 2.1. Materials

#### 2.1.1. Animals

Male C57BL/6N mice, weighing 18 to 20 g, were purchased from Beijing Viton Lihua Laboratory Animal Technology Co., Ltd. (Beijing, China).

All experiments were performed in accordance with the Regulations of the Experimental Animal Administration, issued by the State Committee of Science and Technology of the People’s Republic of China (14 November 1988), the ARRIVE Guidelines, and the Guidelines for Care and Use of Laboratory Animals of the Beijing Institute of Pharmacology and Toxicology. And the experiments were approved by the Animal Ethics Committee of the Beijing Institute of Pharmacology and Toxicology (Protocol code: IACUC-DWZX-2025-P637). SD rats (2-week-old) were obtained from SPF Biotechnology Co., Ltd. (Beijing, China), and Kunming mice (20 ± 2 g) and C57BL/6N (19 ± 2 g) mice were obtained from Beijing Vital River Laboratory Animal Technology Co., Ltd. (Beijing, China). Rodents had free access to sterilized food and distilled water and were maintained in stainless steel cages filled with hardwood chips in an air-conditioned room on a 12:12 h light/dark.

#### 2.1.2. Cell Lines

RAW264.7 and A549 cell lines used in the present study were purchased from the National Infrastructure of Cell Line Resource (Beijing, China).

#### 2.1.3. Reagents

Pentobarbital sodium and lipopolysaccharide (LPS) were procured from Sigma-Aldrich Corp. (St. Louis, MO, USA). The Ultrapure RNA Kit was purchased from Jiangsu Kangwei Century Technology Co., Ltd. (Taizhou, China). The Prime Script RT Reagent Kit and TB Green Premix were obtained from Baori Doctor Biotechnology Co., Ltd. (Beijing, China). Pierce RIPA Buffer, Hoechst, and Dulbecco’s Modified Eagle’s Medium (DMEM) were purchased from Thermo Fisher Scientific Inc. (Waltham, MA, USA). The BCA protein quantification assay kit was obtained from Jiangsu Kaqi Biological Technology Co., Ltd. (Taizhou, China). CCK-8 solution was purchased from Novo Wel Bio-Technology Co., Ltd. (Nanjing, China). Propidium iodide (PI) was procured from Med Chem Express (Monmouth Junction, NJ, USA). The primary antibodies, including anti-interleukin-6 (anti-IL-6), anti-sPLA2, anti-Actin, and anti-rabbit immunoglobulin G (IgG) horseradish peroxidase (HRP)-conjugated secondary antibodies, were purchased from Abcam (Cambridge, UK). Actin Forward 5′-CAGCCTTCCTTCTTGGGTATG-3′; Actin Reverse 5′-GGCATAGAGGTCTTTACGGATG-3′. sPLA2 Forward 5′-TTCAGCGAAGCAACCAGGA-3′; sPLA2 Reverse 5′-CACCAAGGCCACAATAACAGC-3′.

### 2.2. Methods

#### 2.2.1. Construction of LPS Pulmonary Infection Model

To investigate a method for inducing bacterial pneumonia, the tracheal instillation of varying concentrations of LPS was employed, and three LPS concentrations were designed to explore the optimal dosage. According to the principles of randomized grouping design, the mice were divided into four groups that were treated with 0, 0.3, 3, and 15 μg of LPS, respectively, after being anesthetized. A 30 μL volume of LPS liquid with different concentrations was instilled into the trachea using a puncture technique. The group with the symptoms of typical pulmonary infections was selected for subsequent experiments.

#### 2.2.2. Administration and Organ Collection in Mice

In order to investigate the therapeutic effect and the mechanism of the PLA2 inhibitor against bacterial pneumonia, drugs were administered via intraperitoneal injections of varying concentrations to explore the optimal therapeutic dosage. In this experiment, the mice were randomly divided into a control group treated with saline and an experimental group treated with different concentrations of PLA2 inhibitors. All of the mice were injected intraperitoneally with 1 mL of drugs with concentrations of 0, 0.5, and 1.0 mg/mL, respectively. Half an hour later, 30 μL of LPS solution (0.1 mg/mL) was administered to the mice by the puncture technique after being anesthetized. After 24 h, the left lung was collected and soaked in 4% paraformaldehyde in 4 °C.

#### 2.2.3. Western Blot

The lung tissue was ground and stored in RIPA lysis buffer (150 mM NaCl, 50 mM Tris-HCl, pH 7.5, 0.5 M ethylenediamine tetra-acetic acid, Halt protease inhibitor cocktail, and 1% Triton X-100) at low-temperature conditions for 15 min. Then, the samples of lungs were centrifuged at 4 °C and 12,000× *g* for 15 min to obtain the supernatant. The protein concentration of all samples was determined by the BCA method to ensure the consistent protein content for each sample by adding an appropriate amount of buffer solution. After that, the proteins in the sample were heated in a water bath for 10 min to be denatured, and then were separated by 10% polyacrylamide SDS gel electrophoresis. Subsequently, they were transferred onto a PVDF membrane. The membrane was incubated with the primary antibody overnight and then with the secondary antibody for 2 h. Finally, development and photography were carried out in an imaging system. The normalization method was the ratio of the gray value of the target protein to that of Actin in the corresponding sample.

#### 2.2.4. Reverse Transcription Quantitative Real-Time Polymerase Chain Reaction (RT-qPCR)

The total RNA of RAW264.7 cells was extracted using an ultrapure RNA extraction kit, and the RNA concentration was detected using a micro-spectrophotometer. A 20 μL reverse transcription system was prepared, i.e., 2 μL of gDNA remover, 4 μL of 5× reaction mixture, 1 μg of RNA, and RNase-free water was added to obtain a final volume of 20 μL. The reaction conditions were 37 °C for 15 min and 85 °C for 5 s. RT-qPCR detection was performed. A 20 μL reaction system was prepared, i.e., 0.4 μL of each of the upstream and downstream primers, 8.8 μL of RNase-free water, 10 μL of RT-qPCR mixture, and 0.4 μL of cDNA. The reaction conditions were 95 °C for 30 s, 95 °C for 5 s, and 60 °C for 30 s, with a total of 40 cycles. Using Actin as an internal reference, the 2^−ΔΔCt^ method was used to calculate the relative expression level of the target gene mRNA.

#### 2.2.5. Hematoxylin and Eosin (H&E) Staining

The tissue samples were fixed with formalin for organization, followed by a series of wash steps to remove the fixative and so on. The purified tissues were sequentially dehydrated by soaking in ethanol from low to high concentrations. Then, the dehydrated tissues were further permeated with acetone and paraffin sequentially with a constant high temperature to ensure paraffin penetration and totally wrapped for cutting into thin slices of 3 or 5 µm thickness. The tissue slices were stained with hematoxylin dye, for staining the nucleus blue, and eosin dye, for staining the cytoplasm red, after removing the paraffin with a defatting agent. Finally, the stained slices were washed with ethanol from high to low concentrations to remove excess dye and fixed on glass slides and covered with a slip.

#### 2.2.6. Immunofluorescence Staining

The tissue samples were fixed with paraformaldehyde and dehydrated with ethanol to ensure the complete penetration of the antibody and fluorescent label. The specific primary antibody of IL-6 or PLA2 was used to first bind the target protein, and then the unbound antibodies were removed for preventing non-specific binding through repeated rinsing with PBS. The green fluorescently labeled secondary antibody was applied to link fluorophores with a specific primary antibody. The nuclei of samples were stained with the DAPI dye to give blue fluorescence. Finally, the samples were washed multiple times to remove all the unbound antibodies and fixed as described before. The fluorescence microscope was used to observe the sample and excite and detect the fluorescence signal at an appropriate wavelength.

#### 2.2.7. CCK-8

RAW264.7 and A549 were applied and planted in 96-well plates (10^4^ cells/well). After 24 h hours later, cells were incubated with series doses of LPS in each well for 24 h hours. The CCK-8 reagent was added into each well and incubated for 2 h. Cell vitality was tested by the absorbance of each well at a wavelength of 450 nm.

For evaluating the mechanism and therapy effect of PLA2 inhibitors, two PLA2 inhibitors (trans-benzylideneacetone and varespladib) in different concentrations (0 to 80 μg/mL) were added to LPS-stimulated cells. And the cell vitality was tested using the same CCK-8 method as described before.

#### 2.2.8. Flow Cytometry and Confocal Microscopy

RAW264.7 cells were obtained in the logarithmic growth phase with a density of 0.5 × 10^5^ cells/mL and transferred to confocal dishes. Cells were randomly divided into six groups (*n* = 3): controls, untreated LPS stimulation (25 ng/mL LPS), LPS stimulation and treated with PL (25 ng/mL LPS, 0.4 μg/mL PL), LPS stimulation and treated with PL and trans-benzylideneacetone (25 ng/mL LPS, 0.4 μg/mL PL), and LPS stimulation treated with PL and varespladib (25 ng/mL LPS, 0.4 μg/mL PL). The cells were incubated with LPS and inhibitors for 24 h and then stained with the dyes of PI (0.5 mg/mL) and Hoechst (0.5 mg/mL) for 15 min. Cells were dissolved from the culture dish for dispersing into a suspension and characterized by flow cytometer with the excitation and emission wavelengths of PI (ex: 493 nm, em: 636 nm) and Hoechst (ex: 350 nm, em: 460 nm).

### 2.3. Statistical Analysis

For comparisons between two groups, an F-test was first performed to assess the homogeneity of variances, followed by an unpaired *t*-test. When analyzing multiple groups, the Brown–Forsythe test was initially conducted to evaluate variance homogeneity. Subsequently, one-way ANOVA with Tukey’s post hoc analysis was employed. Multiple comparison corrections were implemented using the false discovery rate method. A *p* value of less than 0.05 was considered statistically significant. All data presented as mean ± standard deviation, with results presented up to two decimal places.

## 3. Results

### 3.1. The Impact of LPS and PLA2 on Cells

The cellular interaction between LPS and PLA2 was systematically investigated. The CCK-8 assays revealed no significant cytotoxic effects of LPS (Figure 1A,B) even at 80 µg/mL. Especially, the cell viability of RAW264.7 cells was remarkably increased when stimulated with LPS for 24 h (Figure 1B). The RT-qPCR analysis demonstrated the marked upregulation of PLA2 mRNA expression in LPS-treated RAW264.7 cells (Figure 1C), indicating that LPS stimulation could promote PLA2 secretion at the cellular level (Figure 1C).

The destructive effect of PLA2 on cells under complex conditions was investigated by the method of propidium iodine (PI) staining. In the normal state, PI cannot penetrate the cell membrane to stain the cell nucleus owing to the integrity of cell membrane (Figure 1D). Neither phosphatidylserine (PS) nor LPS alone induced apoptosis, as evidenced by the absence of PI nuclear staining (Figure 1E,F). But if PS was mixed with LPS co-treatment, severe apoptosis was observed with PI-stained red nuclei after 24 h (Figure 1G). This aligns with our previous finding that PLA2 converts PS to lyso-PS [10], which might induce apoptosis in a large proportion of cells (Figure 1H). The damaging effect of various PLA2 subtypes mixed with LPS on A549 cells was also investigated. The most significant cytotoxicity was observed in groups III and V, while group IIA, primarily induced by LPS stimulation, exhibited moderate cytotoxicity. It was notable that minimal impact on cells was observed in group IID, likely attributed to its weak interaction with PS (Appendix A). Furthermore, the cytotoxic effects of lyso-PLs on A549 cells revealed that lyso-PS concentrations exceeding 50 ng induced significant cytotoxicity (Appendix A).

### 3.2. LPS and PLA2 Induce Inflammation In Vivo

Twenty-four hours post intratracheal LPS instillation, severe lung damage with extensive bleeding spots in the lungs could be clearly observed (Figure 2A). The inflammatory response induced by LPS was further investigated in vivo. Twenty-four hours after the administration of 3 μg LPS, significantly elevated pulmonary IL-6 levels were observed in the lungs (Figure 2B,C). However, no lung damage was observed when less than 3 μg LPS was administered. Notably, this threshold dose (3 μg LPS) concurrently induced significant PLA2 upregulation in lung tissue (Figure 2D,E), corroborating our in vitro findings. Histopathological examination further elucidated the PLA2-LPS relationship. H&E-stained sections revealed severe inflammatory infiltration in the alveoli following LPS exposure (Figure 2F). However, with the simultaneous co-administration of LPS and PS, the lung damage worsens, which is proven by the more severe alveolar infiltration (Figure 2F). Immunofluorescence analysis further confirmed strong correlation between the PLA2 overexpression and lung injury severity (Figure 2G). Notably, PLA2 expression was considerably elevated in the LPS+PS group compared to LPS treatment alone. 

### 3.3. PLA2 Inhibitors Attenuate Cellular Damage In Vitro

The effects of four PLA2 inhibitors (Figure 3A) on cell viability were evaluated. The CCK-8 assay results indicated that bromoenol lactone (BL, Figure 3A(I)) and anthranilic acid (ACA, Figure 3A(II)) induced 90% and 40% apoptosis even at a low concentration of 20 µM, respectively. Trans-benzylideneacetone (BZA, Figure 3A(III)) maintained 80% cell viability at a concentration of 40 µM and showed no cytotoxic effects at a concentration of 20 µM. Varespladib (Var, Figure 3A(IV)) showed no stimulatory effects on cells even at a concentration of 40 µM (Figure 3B). However, when treated with the PLA2 inhibitors BZA and Var, the cell retained partial membrane integrity, resulting in diminished PI staining (Figure 3C). Moreover, at the same concentration, Var exhibited better ability for the inhibition of PI staining compared to BZA. Furthermore, quantitative flow cytometric was applied to quantitatively assess the therapeutic effect of inhibitors on cells. The results indicated that LPS treatment alone did not alter membrane stability, preventing PI entry (Figure 3D and Appendix A). However, when cells were co-treated with LPS and PS, the PI fluorescence intensity increased approximately 11-fold compared to the LPS-only and control groups. Following treatment with BZA or Var, the PI fluorescence intensity decreased significantly by ~30%, with no statistically significant difference between the two treated groups.

### 3.4. PLA2 Inhibitors Attenuate LPS Lung Injury in Mice

Mice were administered LPS via tracheal instillation and subsequently treated with BZA (0.5 mg and 1.0 mg) or Var (0.5 mg and 1.0 mg) as illustrated in Figure 4A. Pulmonary IL-6 levels and histopathology were assessed to evaluate the therapy effect of PLA2 inhibitors. In comparison to the controls, the LPS-exposed mice exhibited a notable rise in the expression of IL-6. A single administration of either inhibitor significantly reduced the IL-6 levels in both treatment groups (Figure 4E). Var treatment also restored IL-1β, TNF-α, and MCP-1 concentrations in lung tissue and lung function to levels resembling the unexposed controls (Appendix A).

For further pathological evaluation, lung tissues were examined by H&E staining. Due to the stimulation of LPS, the lungs in the poisoned group exhibited higher levels of red blood cells, increased inflammatory cell infiltration, a less distinct alveolar structure, noticeable congestion, and edema (Figure 4G). However, the lungs in the groups treated with BZT (Figure 4H,I) and Var (Figure 4J,K) displayed no substantial distinction from the control group (Figure 4F), with clear alveolar structure, minimal green fluorescence, and no congestion. Based on the evaluation of sPLA2 immunofluorescence in the same tissues, the green fluorescence from sPLA2 was observed in the lung interstitium and alveolar cavity after being poisoned with LPS (Figure 4G). However, for the groups treated with BZA, the green fluorescence in the lungs significantly reduced. But the fluorescence signal of PLA2 was clearly observed in the lungs even when treated with BZA (1.0 mg) (Figure 4 I,J). Var showed remarkable PLA2-inhibitory effects when treated with Var (1.0 mg), and almost no significant PLA2 fluorescence signal was observed in the lungs, reaching the level of the normal unpoisoned group (Figure 4J,K).

## 4. Discussion

### 4.1. LPS-Induced Lung Pneumonia Models

LPS is a component of the outer membrane of Gram-negative bacteria [11] consisting of a lipid (lipid A) and a polysaccharide chain and acts as an endotoxin by activating the immune system and inducing a strong inflammatory response, which may be caused by its interaction with the receptor called Toll-like receptor 4 (TLR4) present on the surface of the host cell membrane [12]. Therefore, LPS has been widely used to simulate bacterial lung infection and construct a model of lung injury in numerous studies [13]. As the core objective was to investigate the relationship between inflammation and PLA2 triggered by bacteria in the lungs rather than the bacteria themselves, LPS-induced pneumonia models were chosen in this study [14]. Compared to models of lung injury caused by real bacterial infections, the model of lung injury induced by LPS is more stable, reproducible, and easy to operate, facilitating the accurate measurement of biochemical indicators such as inflammatory factors in the later stages.

### 4.2. Non-Irritating Effects of LPS In Vitro

The interaction between LPS and PLA2 was investigated at the cellular level. Initial experiments assessed the cytotoxic effect of LPS on cells. A549 and RAW264.7 cells were utilized in this study as the standard models for pulmonary injury and inflammatory responses [15,16]. It is worth noting that LPS, usually utilized for lung injury models, was almost non-irritating to cells (Figure 1A) and even promoted proliferation (Figure 1B). We discovered in our research that the real damage to cells was caused by PLA2 secreted after LPS stimulation (Figure 1C). Specifically, PLA2 alone showed no cytotoxic effects, and it interacted with phosphatidylserine (PS), a component of cell membrane fragments [17]. PLA2 catalyzed the conversion of PS into lyso-PS and rapidly induced cell apoptosis (Figure 1H). Overall, in vitro, all of the components such as LPS, PLA2, and PS were individually safe for cells. When PLA2 interacted with PS, cellular balance was disrupted due to lyso-PS generation. Apoptotic cells released additional membrane fragments, which were further catalyzed by PLA2 into lyso-PS, amplifying apoptosis (Figure 1I).

### 4.3. LPS Induces Injury In Vivo

In this study, we demonstrated that LPS toxicity exhibits clear dose–response and route-dependent characteristics [18,19,20]. Furthermore, the mechanism of LPS-induced lung injury was further validated in animal models. In contrast to its minimal effects observed in vitro, LPS caused great damage to the lungs in vivo (Figure 2A). The significant difference in damage between the cellular and animal levels indicated that the lung injury caused by LPS was not directly due to LPS destroying lung-related cells, but was through the triggering of other damage pathways in the complex physiological reaction of the body, such as PLA2. This damage process was simulated simply via the pathological evaluation (Figure 2F,G), which showed that LPS caused severe lung damage and stimulated the overexpression of PLA2. The artificial co-administration of LPS with PS to simulate cell membrane fragments exacerbated both lung injury and PLA2 secretion (Figure 2F,G). However, in the absence of PS, as the dose of the toxin LPS increased, PLA2 did not show significant dose dependency (Figure 2D). Overall, LPS did not directly damage cells but acted as a trigger or an initiator. When high PLA2 expression and minimal cell membrane fragmentation occurred, LPS rapidly amplified the damage cascade. Therefore, based on prior investigations examining LPS-induced cytotoxicity in vitro and tissue damage in vivo [21,22], we hypothesized that secondary immune cells and mediators might be key contributors to the LPS-induced PLA2 upregulation. Consistent with this mechanism, our in vitro experiments demonstrated that PLA2 secretion was significantly upregulated (*p* < 0.01) in RAW264.7 macrophages following 24 h of LPS stimulation (Figure 1C).

### 4.4. PLA2 Inhibitors Attenuate Cellular Damage

Given the established association between PLA2 and LPS-induced pulmonary inflammation, the potential therapeutic benefit of PLA2 inhibition was first evaluated at the cellular level using four well-characterized PLA2 inhibitors. Bromoenol lactone (BL, Figure 3A(I)) is an irreversible, selective, and potent inhibitor of calcium-dependent iPLA2 [23], which can inhibit the exocytosis of mast cells stimulated by an antigen without blocking the influx of Ca^2+^. N-(p-amylcinnamoyl) Anthranilic Acid (ACA, Figure 3A(II)), a potential treatment for arrhythmia, is a broad-spectrum PLA2 inhibitor and TRP channel blocker that reversibly inhibits calcium-activated chloride channels [24]. Trans-benzylideneacetone (BZA, Figure 3A(III)) is an immunosuppressant, produced from a metabolite of Gram-negative entomopathogenic bacterium *Xenorhabdus nematophila* [25]. Varespladib (Var, inhibitor IV, Figure 3A(IV)), a selective group IIA sPLA2 inhibitor demonstrating cross-species efficacy in serum models (rat, rabbit, guinea pig, and human) [26], has been clinically validated as safe and well tolerated in multi-dose regimens spanning 12 weeks at maximum daily doses of 1000 mg [27].

The four inhibitors were systematically evaluated both in vitro and in vivo. Cell viability assays revealed significant cytotoxicity for BL and ACA (Figure 3B), leading to their exclusion from further studies. All results demonstrated that PLA2 inhibitors, BZA and Var, effectively alleviated LPS-induced pulmonary inflammation. It was worth noting that, when the activity of PLA2 was effectively suppressed (Figure 4J,K), reducing the series of subsequent damage, the expression of PLA2 enzyme could be inhibited at the source. On the one hand, Var has shown better therapeutic effects against LPS and has been reported as a candidate in the phase III clinical trials for atherosclerosis treatment, demonstrating good safety. On the other hand, BZA has been reported and applied as a drug to suppress insect immune responses. Considering the above reasons, Var should be paid more attention in further research.

## 5. Conclusions

This study revealed that LPS itself does not directly cause pulmonary damage, but ly-PLs, a reaction product of PS and PLA2 secreted after LPS stimulation, is one of the direct causes of severe lung injury. Inhibiting PLA2 can effectively alleviate lung injury induced by LPS, and we have delved into the therapeutic mechanisms associated with it. The mechanism behind the effectiveness of PLA2 inhibition is associated with a reduction in the reaction and products of PLA2 and PS, as well as the release of ly-PLs. These findings establish a mechanistic foundation for developing novel therapies against bacterial pneumonia-associated lung injury and identify Var as a particularly promising therapeutic candidate. 

## Figures and Tables

**Figure 1 cimb-47-00516-f001:**
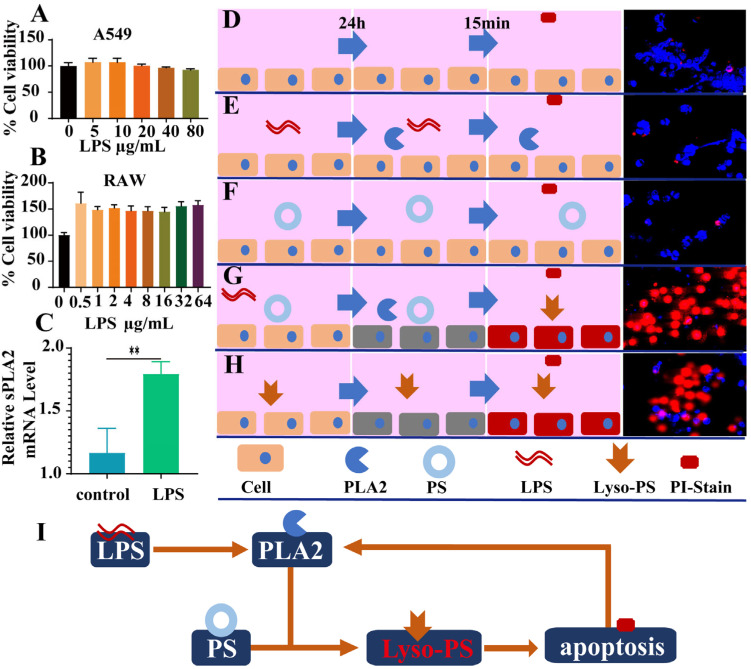
The inductive effect of LPS leads to the generation of phospholipase A2 in cells, and the products generated in the reaction with PL may damage the cells. (**A**) The cell viability of A549 cells (*n* = 6). (**B**) The cell viability of RAW264.7 cells (*n* = 6). (**C**) The expression of sPLA2 mRNA in RAW264.7 cells after 24 h of LPS stimulation (*n* = 4). (**D**–**H**) Confocal microscopy results showing PI staining after 15 min of culturing RAW264.7 cells under normal conditions for 24 h (**D**), with LPS stimulation (**E**), PL (**F**), LPS and PL (**G**), and lyso-PLs for 24 h (**H**). (**I**) The schematic diagram shows the process of LPS stimulating cells to produce phospholipase A2 (** *p* < 0.01).

**Figure 2 cimb-47-00516-f002:**
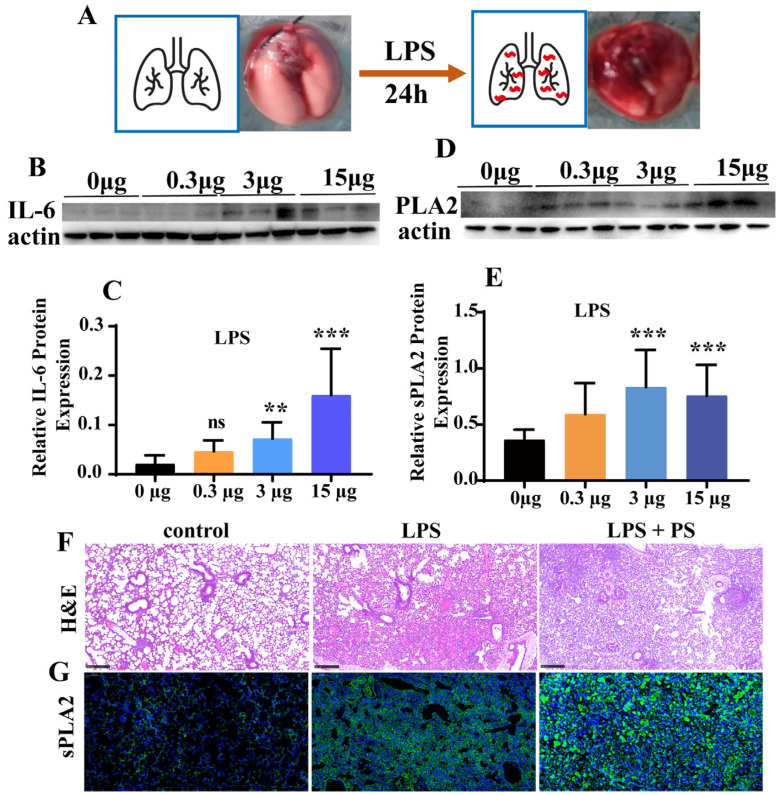
LPS induced inflammation in animals. (**A**) The images reflected the lung injury of mice 24 h after LPS treatment by tracheal instillation. (**B**,**C**) The changes in IL-6 protein expression levels in lung tissue of mice 24 h after LPS treatment by tracheal instillation, *n* = 10. (**D**,**E**) The sPLA2 protein expression levels in the lung tissue of mice 24 h after LPS treatment by tracheal instillation, *n* = 10. (**F**,**G**) Results of lung tissue staining with H&E and sPLA2 immunofluorescence (** *p* < 0.01, *** *p* < 0.001).

**Figure 3 cimb-47-00516-f003:**
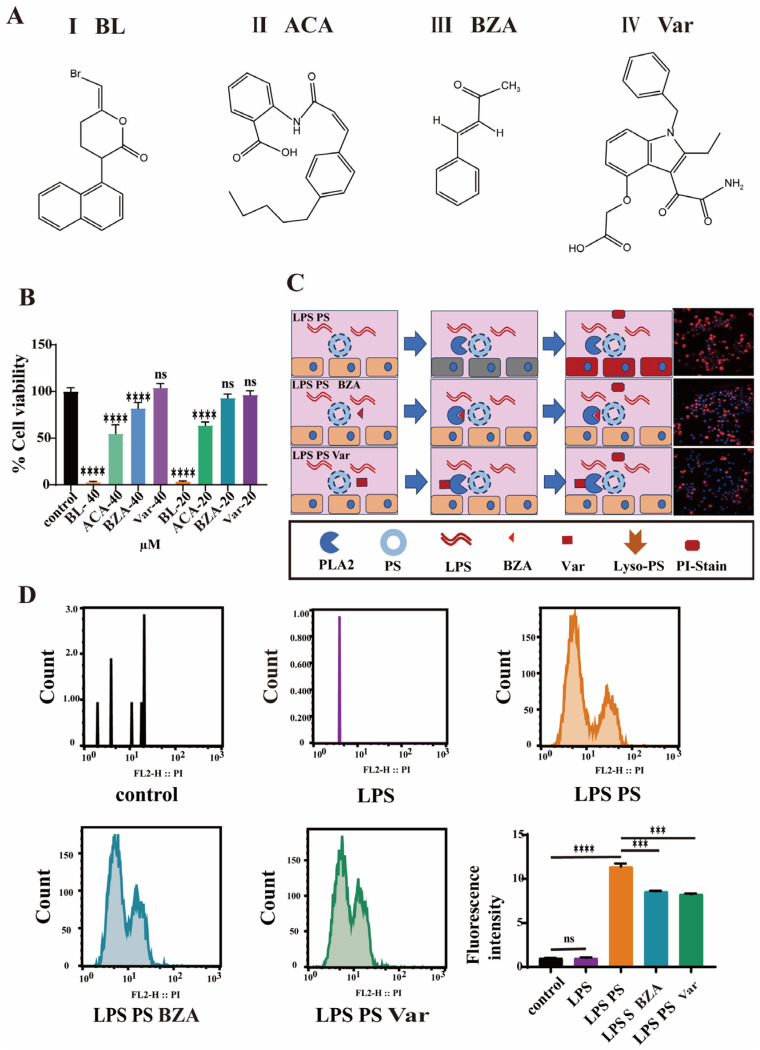
Cellular efficacy assessment of phospholipase A2 inhibitors. (**A**) The chemical formula of PLA2 inhibitors. (**B**) The cell viability of RAW264.7 cells with PLA2 inhibitors (*n* = 6). (**C**) Confocal microscopy results showing RAW264.7 cells stained with PI. (**D**) The results of flow cytometry analysis showing the staining of RAW264.7 cells with PI (*** *p* < 0.001, **** *p* < 0.0001).

**Figure 4 cimb-47-00516-f004:**
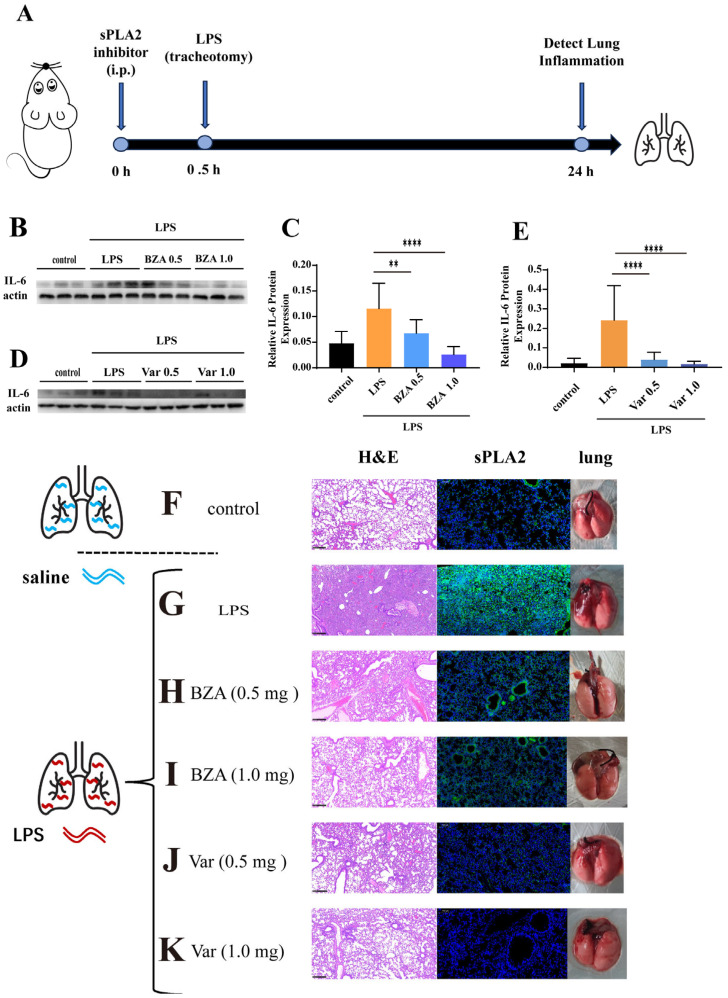
The inhibition assessment of phospholipase A2 in animal models. (**A**) The schematic diagram shows the process of intraperitoneal injection and tracheal instillation of LPS in mice. (**B**–**E**) The graphs show the IL-6 protein expression in lung tissues; *n* = 3 and *n* = 10. (**F**) H&E staining, sPLA2 immunofluorescence staining, and tracheal instillation of saline. (**G**) Results of H&E staining, sPLA2 immunofluorescence staining, and tracheal instillation of LPS. (**H**,**I**) Results of H&E staining, sPLA2 immunofluorescence staining, and tracheal instillation of LPS, following the intraperitoneal injection of varespladib (0.5 mg and 1.0 mg). (**J**,**K**) Results of H&E staining, sPLA2 immunofluorescence staining, and tracheal instillation of LPS, following the intraperitoneal injection of trans-benzylideneacetone (0.5 mg and 1.0 mg) (** *p* < 0.01, **** *p* < 0.0001).

## Data Availability

The data that support the findings of this study are available from the corresponding author upon reasonable request. Images from this study were supported by vecteezy.

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
