# Peer review of "Inhibiting the Interaction Between Phospholipase A2 and Phospholipid Serine as a Potential Therapeutic Method for Pneumonia"

_cimb, 2025, doi:10.3390/cimb47070516_

Round 1
Reviewer 1 Report
Comments and Suggestions for Authors
The manuscript under review describes the use of murine models of deep-lung infection simulation using purified LPS in place of bacteria. Having established that LPS alone does not cause damage to the murine lung, LPS and phosphatidylserine were co-administered, leading to lung damage that was preventable by administration of the phospholipase A inhibitors trans-benzylideneacetone or varespladib.
While the methodology provided is sound, there is no description of the parameters used in mRNA quantification, though the materials used are provided.
The interpretation of the results are fine, with visual representation of what is present at each stage in the form of schematic diagrams. The results appear clear-cut on the whole and have been presented clearly. To further improve presentation of results in Figure 3, it may be beneficial to either remove colour from the bar chart in panel D, or to match the colours to those in the flow cytometry histograms. In the same part of the results, it could also be argued that controls lacking LPS but containing PS should be included - while this control was shown in confocal microscopy images, it is prudent to show that the same holds true in other methodology used within a study.
On line 206, "POPS" is used without having been defined - presumably this was supposed to say "PS"; the same appears in Figure 2. Also in Figure 2, the PLA2 and IL-6 labels in the figure have been swapped in the figure legend.
From the reference list, it appears that the manuscript in its current form has had material removed, as there are more references in the list than have been cited in-text - the citation numbers will therefore need to be carefully checked against those in the reference list..
Overall, this looks like a tidy study with results that benefits the community's understanding of LPS-induced pulmonary damage.
Some aspects of the text require amendment, as both the sentence structure and words used are awkward in places. For example:
which might induce a large number of cell apoptosis → which might induce apoptosis in a large proportion of cells
After poisoned with LPS reaching above 3 µg for 24 h → Twenty-four hours after administration of 3 μg LPS
However, there is no apparent lung damage if the poisoned amount of PLS below 3 µg → However, no lung damage was observed when less than 3 μg LPS was administered
Though these examples are from a single section, similar linguistic difficulties are present throughout much of the manuscript.
Author Response
Dear reviewer,
Sorry, we uploaded the wrong attachment. The correct response is as follows:
1. Point-by-point response to Comments and Suggestions for Authors
Comments 1: While the methodology provided is sound, there is no description of the parameters used in mRNA quantification, though the materials used are provided.
Response 1: We would like to thank the Reviewer for the professional suggestion. I have supplemented the method for mRNA quantification, and the specific content is as follows:
2.2.4. Reverse Transcription Quantitative Real-time Polymerase Chain Reaction (RT-qPCR)
The total RNA of RAW264.7 cells was extracted using an ultra-pure RNA extraction kit, and the RNA concentration was detected using a micro-spectrophotometer. A 20 μL reverse transcription system was prepared: 2 μL of gDNA remover, 4 μL of 5x reaction mixture, 1 μg of RNA, and RNase-free water was added to a final volume of 20 μL. The reaction conditions were 37 °C for 15 min and 85°C for 5 s. RT-qPCR detection was performed. A 20 μL reaction system was prepared: 0.4 μL of each of the upstream and downstream primers, 8.8 μL of RNase-free water, 10 μL of RT-qPCR mixture, and 0.4 μL of cDNA. The reaction conditions were 95 °C for 30 s, 95 °C for 5 s, and 60 °C for 30 s, with a total of 40 cycles. Using GAPDH as an internal reference, the 2-ΔΔCt method was used to calculate the relative expression level of the target gene mRNA.
The specific revisions are on page 3, paragraph 7-8, lines 120-131 of the manuscript.
Comments 2: The interpretation of the results are fine, with visual representation of what is present at each stage in the form of schematic diagrams. The results appear clear-cut on the whole and have been presented clearly. To further improve presentation of results in Figure 3, it may be beneficial to either remove colour from the bar chart in panel D, or to match the colours to those in the flow cytometry histograms. In the same part of the results, it could also be argued that controls lacking LPS but containing PS should be included - while this control was shown in confocal microscopy images, it is prudent to show that the same holds true in other methodology used within a study.
Response 2: Thank you very much for the suggestions of the Reviewer. I have unified the colors of the bar charts for each group in Figure 3D with those of the flow cytometry histograms, and the specific content is as follows:
The specific revisions are on page 7, paragraph 2, lines 250 of the manuscript.
Further addressing the reviewer’s comment, we have supplemented the flow cytometry data using PS as a control group. The results demonstrate no significant difference in PI staining between PS-treated cells and untreated controls, supporting the relative biosafety of the PS material. These data have been included in the Supplementary Information of the manuscript.
The specific revisions are on page 7, paragraph 1, lines 245 of the manuscript.
Comments 3: On line 206, "POPS" is used without having been defined - presumably this was supposed to say "PS"; the same appears in Figure 2. Also in Figure 2, the PLA2 and IL-6 labels in the figure have been swapped in the figure legend.
Response 3: We would like to thank the Reviewer for their expert criticism and offer our sincere gratitude for pointing out this error in our report. I have changed the incorrectly written "POPS" to "PS" in the manuscript and swapped the labels of PLA2 and IL-6 in the legend of Figure 2.
The specific revisions are on page 6, paragraph 2-3, lines 221 and 226 of the manuscript.
Comments 4: From the reference list, it appears that the manuscript in its current form has had material removed, as there are more references in the list than have been cited in-text - the citation numbers will therefore need to be carefully checked against those in the reference list.
Response 4: Thank the reviewers for their comments on my citation of references. I have reorganized the references in the text and re - inserted them into the manuscript.
The specific revisions are on page 12, paragraph 4, lines 398-446 of the manuscript.
2. Response to Comments on the Quality of English Language
Comments 1: which might induce a large number of cell apoptosis → which might induce apoptosis in a large proportion of cells
Response 1: Thank the Reviewer for correcting the sentence structure and the words used in my manuscript, and the specific revisions are on page 5, paragraph 5, lines 195-196 of the manuscript.
Comments 2: After poisoned with LPS reaching above 3 µg for 24 h → Twenty-four hours after administration of 3 μg LPS
Response 2: Thank the Reviewer for correcting the sentence structure and the words used in my manuscript, and the specific revisions are on page 6, paragraph 2, lines 214-215 of the manuscript.
Comments 3: However, there is no apparent lung damage if the poisoned amount of PLS below 3 µg → However, no lung damage was observed when less than 3 μg LPS was administered
Response 3: Thank the Reviewer for correcting the sentence structure and the words used in my manuscript, and the specific revisions are on page 6, paragraph 2, lines 216-217 of the manuscript.

Reviewer 2 Report
Comments and Suggestions for Authors
The purpose of the manuscript "Inhibiting the Interaction Between Phospholipase A2 and Phospholipid Serine as a Potential Therapeutic Method for Pneumonia" is to examine how phospholipase A2 (PLA2) contributes to lung damage caused by LPS and assess the potential of PLA2 inhibitors as a treatment for pneumonia-related inflammation and tissue damage. This study proposes a novel therapeutic strategy utilizing PLA2 inhibitors such as varespladib and outlines a promising mechanism involving PLA2 in pneumonia pathogenesis. One of the strengths is the new theory that connects lung damage and surfactant disruption to PLA2 activity. The therapeutic potential of PLA2 inhibitors, particularly varespladib, is demonstrated. These are a few points:
- There is no pharmacokinetic or dose-response data available for inhibitors or LPS. Consult earlier studies or include pilot toxicity/dose-range studies.
- Control groups differ from experiment to experiment; normalization is not always consistent (e.g., GAPDH). Note normalization techniques and standardize controls.
- Lack of more comprehensive cytokine profiling; excessive focus on IL-6 and H&E. Aim to incorporate functional lung tests, MCP-1, IL-1β, and TNF-α.
- Parametric testing without first establishing equality of variances or normality. When appropriate, employ non-parametric testing in addition to the Shapiro-Wilk test.
- Type I error risk from repeated comparisons without adjustment. After an ANOVA, you can use the FDR or Bonferroni corrections.
- SD/SEM and confidence intervals are absent from several figures. Always indicate the type of variability and include error bars.
- LPS does not show cytotoxicity in vitro, yet it causes damage in vivo. Please look at the functions of secondary immune cells and mediators.
- The measurement of sPLA2 did not differentiate between certain isoforms (e.g., Group IIA, V, X). Please use isoform-specific tests and discuss the significance of subtypes.
- Lyso-PS is believed to mediate apoptosis, even though it cannot be quantified directly. Please measure lyso-PS levels and prevent its effects to confirm causality.
- Minor typographical errors must be revised; the English language needs more improvement.
Round 2
Reviewer 2 Report
Comments and Suggestions for Authors
After thoroughly reviewing the revised manuscript and considering the authors' revisions and responses to the referee's comments, I find that the manuscript has been significantly improved. The authors have effectively addressed the concerns, thereby enhancing the clarity and scientific rigour of their study. The revisions have clarified the methodology, improved the presentation of results, and strengthened the discussion and conclusions.
Therefore, I believe that the manuscript now meets the standards required for publication in CIMB, and I recommend that it be accepted for publication.
Thank you for considering my recommendation.